# Photoluminescent and Scintillating Performance of Eu^3+^-Doped Boroaluminosilicate Glass Scintillators

**DOI:** 10.3390/ma16134711

**Published:** 2023-06-29

**Authors:** Yujia Gong, Lianjie Li, Junyu Chen, Hai Guo

**Affiliations:** Department of Physics, Zhejiang Normal University, Jinhua 321004, China

**Keywords:** scintillators, Eu^3+^, X-ray imaging, glass

## Abstract

In comparison with single crystal scintillators, glass scintillators are more promising materials for their benefits of easy preparation, low cost, controllable size, and large-scale manufacture. The emission of Eu^3+^ ion at 612 nm matches well with the photoelectric detector, making it suitable for the activator in glass scintillators. Therefore, the research on Eu^3+^ doped glass scintillators attract our attention. The photoluminescent and scintillating properties of Eu^3+^-activated boroaluminosilicate glass scintillators prepared by the conventional melt-quenching method were investigated in this work. The glass samples present good internal quantum yield. Under X-ray radiation, the optimal sample reveals high X-ray excited luminesce (XEL), and its integrated intensity of XEL is 22.7% of that of commercial crystal scintillator Bi_4_Ge_3_O_12_. Furthermore, the optimal specimen possesses a spatial resolution of 14 lp/mm in X-ray imaging. These results suggest that Eu^3+^-doped boroaluminosilicate glass is expected to be applied in X-ray imaging.

## 1. Introduction

X-ray is regarded as an excellent radiation source due to its short wavelength and strong penetration and is widely used in structural analysis, perspective imaging, constituent analysis, and other fields. Perspective imaging technology can be applied in medical diagnosis, industrial nondestructive inspection, national security checks, and so on. Scintillators, an important component in perspective imaging technology, absorb the transmitted X-rays and convert them into visible light, thus recording the structural information of the imaging objects. Currently, scintillator materials used in perspective imaging systems are mostly single crystal scintillators, like Bi_4_Ge_3_O_12_ (BGO), (Lu,Y)_2_SiO_5_:Ce, andCsI:Tl [1,2,3,4,5]. High production cost, complex production process, and limited product size confine the wide application of single crystal scintillators [6,7]. Therefore, many new scintillators have been developed continuously. Among them, glass scintillators stand out because of their high transmittance, strong plasticity, and diverse composition [8,9,10].

Glass scintillators consist of glass hosts and luminescent centers. The glass hosts determine the basic physical properties of glass scintillators and provide an appropriate crystal field environment for luminescent centers. Silicate glasses have high mechanical strength and excellent physical–chemical stability. However, the solubility of rare-earth ions in silicate glasses is not ideal, and the melting point is high. Borosilicate glasses can be obtained by adding the proper amount of boron oxide into silicate glasses. Compared with silicate glasses, the melting point of borosilicate glasses decreases obviously, and the solubility of rare-earth ions increases effectively [11]. The addition of alumina makes the glass structure tighter. The addition of yttrium oxide can improve the density and X-ray absorption capacity of the glasses [12]. Therefore, in this paper, SiO_2_-B_2_O_3_-Al_2_O_3_-Y_2_O_3_ glass was selected as a glass host for glass scintillators.

Generally, in glasses, Eu^3+^ will occupy an asymmetric crystal field environment. Its strongest emission peak locates at 612 nm (^5^D_0_ to ^7^F_2_), and matches well with the photoelectric detector. Currently, some works of glass scintillators with Eu^3+^ doping have been reported [13]. Eu^3+^ doped tellurite glass scintillators were reported by Huang et al. [14]. The intergrated intensity of X-ray excited luminesce (XEL) of the optimal sample reached 6% of that of BGO, similarly thereafter. Wantana et al. doped Eu^3+^ into borotungstate glasses and borosilicate glasses as luminescent centers, and the XEL intensity was 8.87% and 13% [15,16], respectively. Guo et al. reported Eu^3+^-doped boroaluminate glass scintillators and the maximum XEL intensity was 18.4% [17]. The performance of Eu^3+^-doped glass scintillators needs to be further improved. In addition, there are no reports on Eu^3+^-doped glass scintillators for X-ray perspective imaging technology.

In this work, a series of Eu^3+^ doped borosilicate glasses with the addition of aluminium oxide and yttrium oxide was fabricated. Transmittances of samples are about 80% at 600 nm, and the average lifetimes of the ^5^D_0_ state of Eu^3+^ are about 1.7 ms. The optimal sample doped with 8 mol% Eu^3+^ exhibited the highest integrated XEL intensity (22.7% of BGO). What is more, for X-ray imaging, its spatial resolution reaches up to 14 lp/mm. Results suggest that boroaluminosilicate glass scintillators doped with Eu^3+^ may have potential use in X-ray imaging.

## 2. Materials and Methods

### 2.1. Preparation

A series of glasses with a nominal composition of 35.4SiO_2_-21B_2_O_3_-13Al_2_O_3_-(30.6−*x*)Y_2_O_3_-*x*Eu_2_O_3_ (mol%, *x* = 2, 4, 6, and 8, and labeled as SBAY:*x*Eu) were fabricated via the melt-quenching method. Raw materials, containing SiO_2_, H_3_BO_3_, and Al_2_O_3_ with A.R. purity, Y_2_O_3_ and Eu_2_O_3_ with 99.99% purity, were weighed by stoichiometric proportions and mixed uniformly in an agate mortar. The mixture was sintered for 1 h at 1500 °C. Then, the melt was pressed using a heated steel plate after being quickly poured over a stainless-steel plate that had been preheated to 300 °C. After cooling to room temperature, these glass samples were put in the furnace and annealed at 800 °C for 3 h to reduce stress. Ultimately, all glass specimens were cut and polished to 2 mm for the following characterization.

### 2.2. Characterization

X-ray diffraction (XRD) patterns of samples were gauged by Rigaku MiniFlex/600 XRD (Tokyo, Japan, CuK*_α_*_1_, λ = 0.154056 nm) equipment. Fourier transforms infrared (FT-IR) spectra was characterized by a NEXUS 670 spectrophotometer (Thermo Nicolet, Waltham, MA, USA). U-3900 spectrophotometer (Hitachi, Tokyo, Japan) was used to test transmission spectra. Excitation and emission spectra, internal quantum efficiency (IQE), and decay curves were investigated with a FS5 spectrofluorometer (Edinburgh Instruments, Livingston, UK) equipped with a 150 W Xe lamp. XEL spectra were accomplished on an OmniFluo960-X-ray scintillator fluorescence spectrometer (Zolix Instruments, Beijing, China). Pictures of X-ray imaging were taken by a Canon camera (EOS600D).

## 3. Results and Discussion

### 3.1. Structural Characters

XRD patterns of glass samples are shown in Figure 1a. No obvious diffraction peak can be observed in the range of 10–70°. All samples present moderate bands which indicate that all samples are glasses instead of glass ceramics.

Figure 1b gives FT-IR spectra of the SBAY host and SBAY:8Eu specimen. The origin of the absorption band around 472 cm^−1^ is bending vibrations of Si-O-Si and O-Si-O [18,19]. The absorption band at 711 cm^−1^ is caused by B-O-B linkage bending vibrations in [BO_3_] groups [12]. The absorption peak at 985 cm^−1^ is attributed to the antisymmetric stretching vibration of Si-O in [SiO_4_] tetrahedra [18]. Furthermore, the asymmetric stretching vibration of B-O of [BO_3_] units brings on the absorption bands located at 1243 and 1375 cm^−1^ [20]. The two curves are relatively similar and have the same absorption peak, indicating that the incorporation of Eu^3+^ does not impact glasses structures obviously.

Figure 2 demonstrates the transmittance spectra of the SBAY host and SBAY:*x*Eu specimens. Absorption peaks at 362, 378, 395, 414, 465, 531, and 579 nm are observed, which correspond to characteristic transitions from ^7^F_0_ to ^5^D_4_, ^5^G_3_, ^5^L_6_, ^5^D_3_, ^5^D_2_, ^5^D_1_, and ^5^D_0_ of Eu^3+^, respectively. Importantly, transmittances of all Eu^3+^-doped specimens are about 80% at 600 nm. As the concentration of Eu^3+^ ions rises, the intensities of all absorption peaks also increase. In addition, the absorption edge also exhibits a significant red shift with increasing Eu^3+^ content. Such phenomenon might be attributed to new unoccupied electron states in the gap below the conduction band edge due to the substitution of Y_2_O_3_ by Eu_2_O_3_ [14,21,22].

On the basis of Beer–Lambert law (*I* = *I*_0_*e*^−*αd*^, here *I* and *I*_0_ are the intensities of the transmitted light and incident light, respectively, *d* is the thickness and *α* is absorption coefficient of samples), the equation of *αd* = −ln *T* can be obtained. The equation *α*^2^ = *B*(*hν* − *E*_g_) (*B* is a constant coefficient, *hν* is the energy of incident photons and *E*_g_ is band gap energy) is applied for the direct band material [23]. So, *E*_g_ value of the SBAY host is estimated to be 3.55 eV.

### 3.2. Photoluminescent Properties

The photoluminescent emission (PL) and photoluminescent excitation (PLE) spectra of SBAY:*x*Eu specimens are displayed in Figure 3a,b, respectively. Excited by 464 nm, five characteristic emission peaks of Eu^3+^ at 579, 593, 612, 653, and 700 nm correspond to transitions from ^5^D_0_ to ^7^F_J_ (J = 0–4) [17,24], respectively. Thereinto, the ^5^D_0_→^7^F_1_ (593 nm) transition of Eu^3+^ is a magnetic dipole transition. It is not affected by the environment of Eu^3+^ because of the selection rule (ΔJ = 1). Furthermore, the transition of ^5^D_0_→^7^F_2_ of Eu^3+^ at 612 nm is an electric dipole transition [16], which is strongly dependent on the symmetry of the environmental structure of Eu^3+^. As demonstrated in Figure 3, the intensity of the emission peak at 612 nm is the strongest. Therefore, Eu^3+^ ions located at an asymmetric crystal field environment. In order to intuitively get the symmetry of the environment of Eu^3+^, the integral intensity ratio (*R*) is calculated by the following formula [25],
(1)$$R=I_{ED}/I_{MD},$$
where *I*_ED_ is luminescent intensity of electric dipole transition, and *I*_MD_ is luminescent intensity of magnetic dipole transition. The values of *R* of SBAY:*x*Eu are listed in Table 1. The values of *R* decrease slightly with increasing Eu^3+^ concentration, indicating that environmental structure of Eu^3+^ becomes more ordered slightly.

In addition, excitation peaks at 318, 362, 383, 393, 413, 464, 531, and 579 nm can also be observed in PLE spectra (λ_em_ = 612 nm) in Figure 3b, which corresponds to the characteristic transitions from ^7^F_0_ to ^5^H_3_, ^5^D_4_, ^5^G_3_, ^5^L_6_, and ^5^D_3, 2, 1, 0_ of Eu^3+^ [26,27], respectively. The excitation band at about 250–310 nm is attributed to the charge transfer (CT) band [17].

Both PL and PLE spectra show that the luminescent intensity of Eu^3+^ increases first and then decrease with increasing Eu^3+^ concentration, and the optimal sample is SBAY:6Eu. The reason is that with increasing concentration of Eu^3+^, the concentration quenching phenomenon occurs because the possibility of non-radiation transition is promoted.

The values of IQE were measured to evaluate the optical performance of glass samples and computed by the equation followed [28],
(2)$$\mathrm{IQE}=\int L_{\mathrm{specimen}}/\left(\int E_{\mathrm{reference}}-\int E_{\mathrm{specimen}}\right),$$
where *L*_specimen_ is the emission intensity of specimen, *E*_reference_ and *E*_specimen_ are excitation intensities with BaSO_4_ and specimen, respectively. The corresponding spectra are demonstrated in Figure 4 and the maximum IQE value is 81.5% for SBAY:2Eu, which is higher than most Eu^3+^-doped glasses [26]. Furthermore, the IQE value of SBAY:6Eu sample is 67.5%, as listed in Table 1.

Figure 5 presents the fluorescent lifetime curves of 612 nm emission of SBAY:*x*Eu (λ_ex_ = 464 nm). Average lifetimes (*τ*) of ^5^D_0_ level of Eu^3+^ were calculated by formula [17,27],
(3)$$\tau =\int_{0}^{\infty }tI_{t}dt/\int_{0}^{\infty }I_{t}dt,$$
where *I_t_* stands for the emission intensity of SBAY:*x*Eu samples at *t* time. As shown in Table 1, as the content of Eu^3+^ increases, the average lifetimes are 1.71, 1.64, 1.71, and 1.43 ms, respectively.

### 3.3. Scintillating Properties

The outstanding transparency, high IQE, pure red emission, and suitable lifetime indicate that Eu^3+^-doped boroaluminosilicate glasses are promising scintillators for X-ray imaging. XEL spectra of SBAY:*x*Eu and BGO are measured to excavate the scintillating performance, as displayed in Figure 6. Peaks at 579, 593, 612, 653, and 701 nm are assigned to transitions of ^5^D_0_→^7^F_J_ (J = 0–4) of Eu^3+^, which are similar to PL spectra. Integrated XEL intensities of samples enhance as increasing concentration of Eu^3+^. The highest integrated XEL intensity is 22.7% (from SBAY:8Eu) of that of BGO. It is higher than other Eu^3+^-doped borate and germanate glass scintillators listed in Table 2.

Relating scintillating mechanism is shown in Figure 7 and described as follows [31,32]. At the first conversion stage, heavy atoms in the host interaction with X-ray irradiation, and, therefore, many electrons and holes are created by the photoelectric effect or Compton scattering. Subsequently, electrons and holes are thermalized to secondary electrons and deep holes [33]. At the second transport stage, secondary electrons and low energy holes are gradually migrated to the bottom of the conduction band and the top of the valence band with the production of phonons, respectively. After that, the luminescent centers (Eu^3+^ ions) absorb the energy of electron-hole pairs and jump to an excited state from the ground state. At the last luminescence stage, Eu^3+^ ions in excited state return to the ground state with the desired scintillation light.

To investigate the radiation tolerance, the optimal SBAY:8Eu specimen was radiated continuously for 60 min by X-ray (6 W). The XEL spectra were measured at five-minute intervals and are given in Figure 8a. Because of some fluctuations in X-ray, the line of integrated XEL intensity is not a straight line (Figure 8b). But, the magnitude of the changes can be almost negligible. Therefore, the specimen has competent radiation tolerance [34].

Furthermore, the resistance under X-ray radiation of the SBAY:8Eu sample is assessed with different input power likewise. As demonstrated in Figure 9a, the XEL intensity of the SBAY:8Eu sample increases with the growth of input X-ray power. Firstly, the transmittance of the SBAY:8Eu sample is higher than 80%. Even though the input power of the X-ray increases from 4 W to 12 W, the transmittance of the SBAY:8Eu sample is steady (Figure 9b). The above results illustrate that Eu^3+^-doped boroaluminosilicate glass possesses pretty good radiation resistance [29].

X-ray imaging ability of Eu^3+^-doped glass scintillators was first reported in this paper (SBAY:8Eu sample). To appraise the practicality of SBAY:8Eu glass for X-ray imaging, the bright field photos of chip, metallic spring in capsule, and standard X-ray imaging test-pattern plate are exhibited in Figure 10a,c,e, respectively. Their X-ray images with high resolution are displayed in Figure 10b,d,f, respectively. And the internal structures of the electronic chip and encapsulated metallic spring can be clearly visualized using an X-ray imaging instrument. As demonstrated in Figure 10f, the spatial resolution of 14 lp/mm can be achieved for the SBAY:8Eu sample. Therefore, the SBAY:8Eu sample with high integrated XEL intensity and excellent spatial resolution (14 lp/mm) might have potential application for X-ray imaging [35].

## 4. Conclusions

A series of Eu^3+^-doped SiO_2_-B_2_O_3_-Al_2_O_3_-Y_2_O_3_ glasses were manufactured by the melt quenching method. All specimens present good optical transmittance and good internal quantum yield. The optimal SBAY:8Eu sample reveals a fine X-ray conversion ability (integrated intensity of XEL is 22.7% of that of BGO crystals), excellent radiation tolerance, and good spatial resolution of 14 lp/mm. Such results suggest that Eu^3+^-doped boroaluminosilicate might be utilized in X-ray imaging.

## Figures and Tables

**Figure 1 materials-16-04711-f001:**
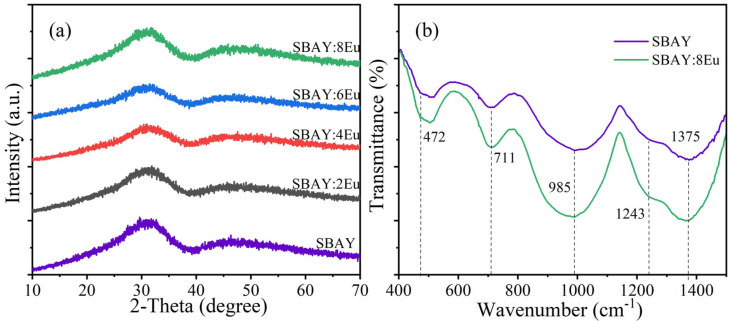
(**a**) XRD patterns of SBAY host and SBAY:*x*Eu specimens, (**b**) FT-IR spectra of SBAY host and SBAY:8Eu specimens.

**Figure 2 materials-16-04711-f002:**
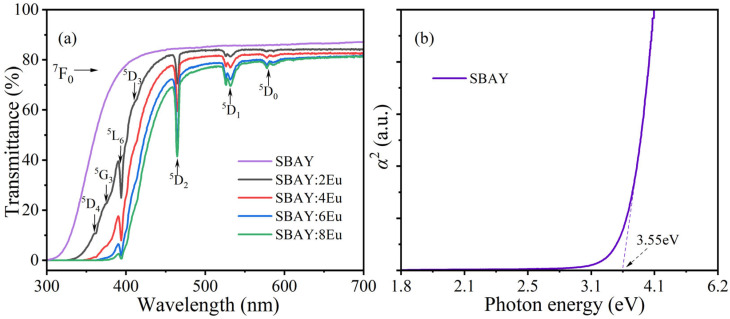
(**a**) Transmittance spectra of SBAY host and SBAY:*x*Eu specimens, (**b**) the relationship of *α*^2^ with photon energy for SBAY host.

**Figure 3 materials-16-04711-f003:**
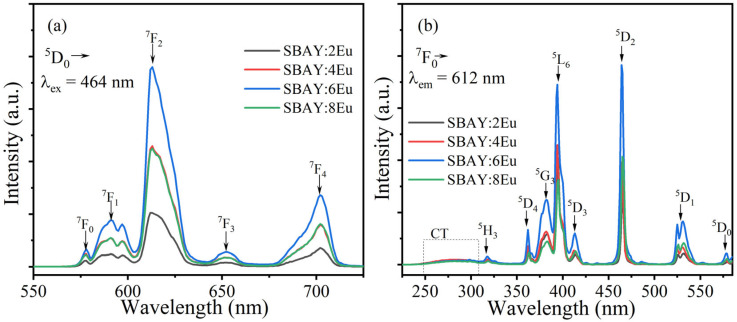
(**a**) PL spectra (λ_ex_ = 464 nm), (**b**) PLE spectra (λ_em_ = 612 nm) of SBAY:*x*Eu specimens.

**Figure 4 materials-16-04711-f004:**
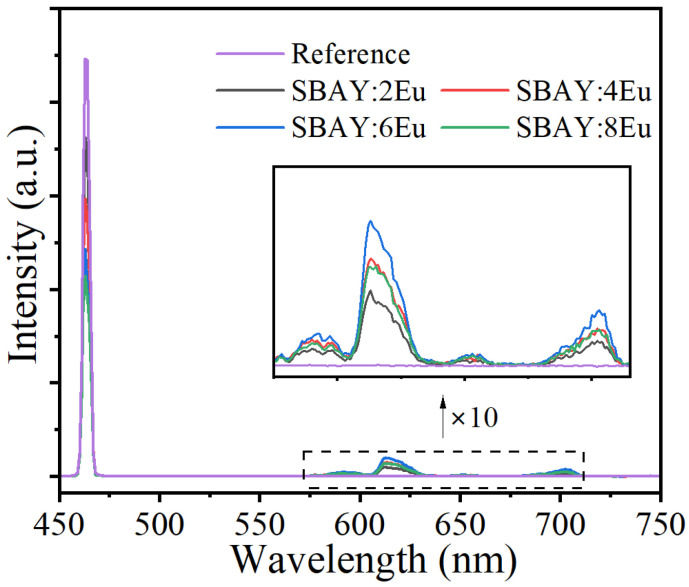
Excitation lines of BaSO_4_ reference, and PL (λ_ex_ = 464 nm) spectra of SBAY:*x*Eu. The inset shows the magnification of PL spectra of SBAY:*x*Eu.

**Figure 5 materials-16-04711-f005:**
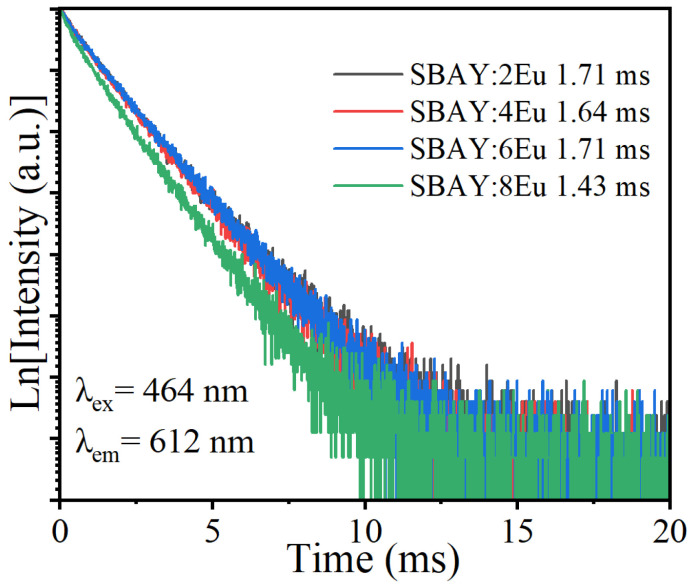
Fluorescent lifetime curves of 612 nm emission of SBAY:*x*Eu specimens.

**Figure 6 materials-16-04711-f006:**
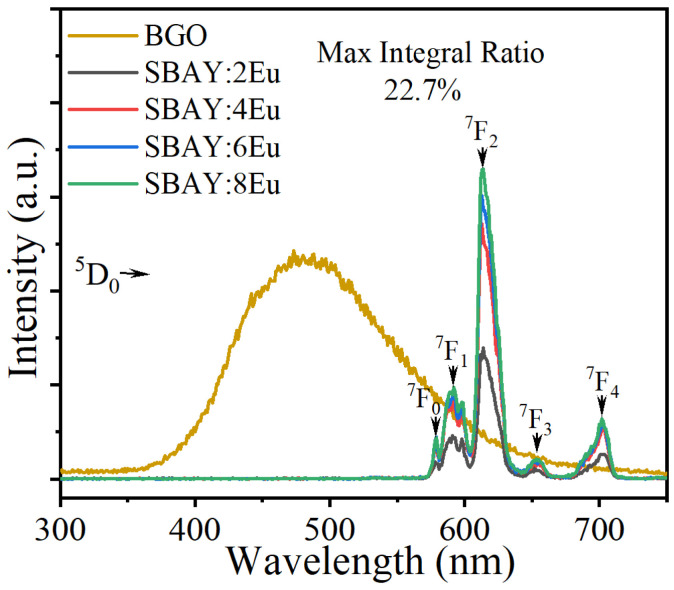
XEL spectra of SBAY:*x*Eu specimens and BGO crystal with different Eu^3+^ content.

**Figure 7 materials-16-04711-f007:**
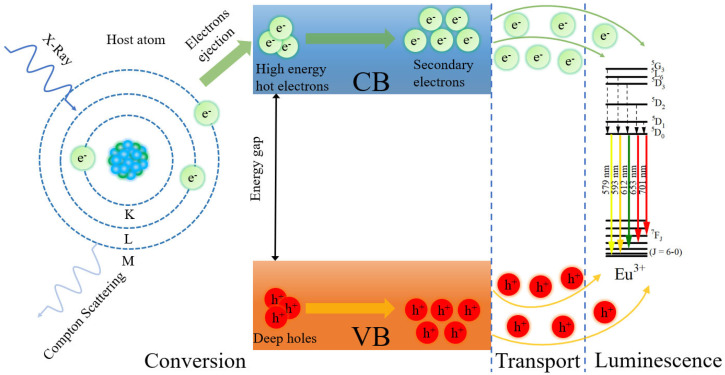
XEL mechanism diagram for SBAY:*x*Eu samples, here CB and VB are conduction band and valence band, respectively.

**Figure 8 materials-16-04711-f008:**
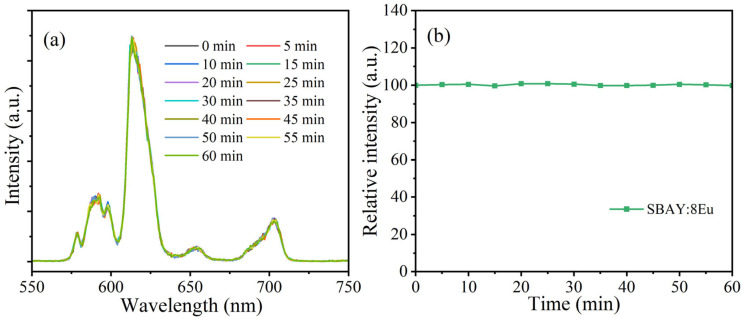
(**a**) The XEL spectra (measured at five-minute interval) of SBAY:8Eu specimen radiated continuously for 60 min by X-ray (6 W), and (**b**) time-dependent integrated XEL intensities for SBAY:8Eu specimen.

**Figure 9 materials-16-04711-f009:**
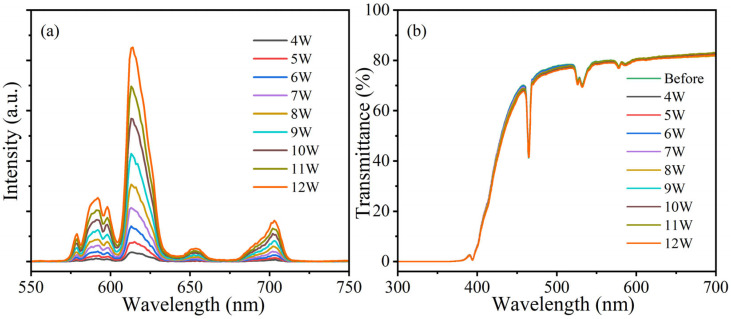
(**a**) XEL spectra of SBAY:8Eu specimen under X-ray radiation with different input power, (**b**) transmittance spectra of SBAY:8Eu specimen after X-ray irradiation with different input power.

**Figure 10 materials-16-04711-f010:**
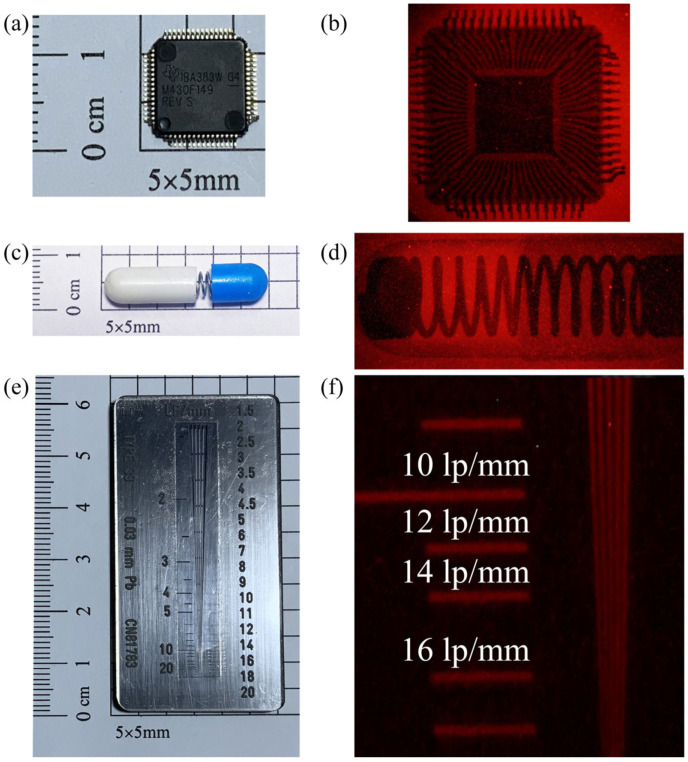
Photos of chip (**a**), metallic spring in capsule (**c**), and standard X-ray test pattern plate (**e**) under daylight. X-ray images of chip (**b**), metallic spring in capsule (**d**), and standard X-ray test pattern plate (**f**) based on SBAY:8Eu.

**Table 1 materials-16-04711-t001:** The values of *R*, IQE (λ_ex_ = 464 nm), fluorescent lifetimes of ^5^D_0_ level of Eu^3+^ and XEL intensity of SBAY:*x*Eu specimens.

Specimens	SBAY:2Eu	SBAY:4Eu	SBAY:6Eu	SBAY:8Eu
*R*	4.06	3.91	3.90	3.89
IQE (λ_ex_ = 464 nm)	81.5%	73.9%	67.5%	43.8%
average lifetime (ms)	1.71	1.64	1.71	1.43
XEL intensity	10.2%	18.8%	21.0%	22.7%

**Table 2 materials-16-04711-t002:** The XEL properties of reported Eu^3+^-doped glass scintillators.

Sample	Material	XEL Properties	Ref.
BGO	Single-crystal	100%	[29]
W_2_O_3_-Gd_2_O_3_-B_2_O_3_	Glass	8.87%	[16]
CaO-Gd_2_O_3_-SiO_2_-B_2_O_3_	Glass	13%	[15]
Al_2_O_3_-B_2_O_3_-Gd_2_O_3_	Glass	18.4%	[17]
GeO_2_-Al_2_O_3_-Na_2_O-LiF-LaF_3_	Glass-ceramic	20%	[27]
Al_2_O_3_-B_2_O_3_-SiO_2_-Y_2_O_3_	Glass	22.7%	This work
ZnO-Ga_2_O_3_-GeO_2_-K_2_O-Al_2_O_3_	Glass	26.5%	[30]

## Data Availability

The data presented in this study are available on request from the corresponding author.

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
