# Peer review of "Photoluminescent and Scintillating Performance of Eu3+-Doped Boroaluminosilicate Glass Scintillators"

_materials, 2023, doi:10.3390/ma16134711_

Round 1
Reviewer 1 Report
This manuscript presents the production of Eu3+ activated SiO2-B2O3-Al2O3-Y2O3 glasses as scintillator materials. This development aim is to design and produce more available scintillators for X-ray imaging. The results presented show the optical transmittance, and good internal quantum yield was reported. The findings were compared to the BGO scintillator.
The data have been appropriately listed in tables, and the figures are precise and well-supported by the text. Statistical checks as necessary to ensure the accuracy of the findings were conducted. The conclusion effectively summarizes the results and discussions, providing a clear and concise summary of the findings.
Major change is required:
Neutron detection using scintillators is widely used in various fields, including nuclear physics, radiation monitoring, and homeland security. Neutron detection, thermal or fast, can sometimes be performed with scintillators. The development of advanced scintillator materials for neutron detection systems, enables their widespread use in diverse scientific and practical applications.
Please add a discussion paragraph regarding the ability to use the Eu3+ activated SiO2-B2O3-Al2O3-Y2O3 glasses for neutron detection.
Moderate editing of English language required.
Author Response
Please see the attachment. The responses to four revierwers are shown in one file to every reviewer.

Reviewer 2 Report
Gong et al. studied the photoluminescence and scintillating performance of Eu3+ doped boro aluminosilicate glass scintillators, preparing them by using a conventional melt-quenching method. Under X-ray radiation, their Eu3+ doped boroaluminosilicate sample reveals high X-ray excited luminesce with the integrated intensity of 22.7% to that of commercial Bi4Ge3O12 crystal scintillator, excellent radiation tolerance, and good spatial resolution of 14 lp/mm suggesting that Eu3+ doped boroaluminosilicate can be utilized in X-ray imaging. There are two sentences in the manuscript (see the attached) where the authors have claimed to be the first to study Eu3+ doped glass scintillators for X-ray imaging. The authors may be unaware, but the claim needs to be corrected. Please fix this in your revision. In addition, the manuscript has a few typographical/grammatical errors. Otherwise, the manuscript is worth publishing after the above-mentioned corrections.

The manuscript has a few typographical and grammatical errors. Please see the attached for a few of them.
Author Response

(The authors gave the same response as above.)

Reviewer 3 Report
The paper of YuJia Gong, LianJie Li, JunYu Chen and Hai Guo “Photoluminescent and scintillating performance of Eu3+-doped boroaluminosilicate glass scintillators” is devoted to study of photoluminescent and scintillation properties of activated Eu3+ scintillators from boroaluminosilicate glass, prepared by melt-hardening method.
I think the paper is of interest for the journal and can be published without substantial changes. However, I can not judge about the quality of the English, but for me some parts of the article sound strange. Although the meaning is clear.
Author Response

(The authors gave the same response as above.)

Reviewer 4 Report
The presented article ‘Photoluminescent and scintillating performance of Eu3+-doped boroaluminosilicate glass scintillators’ is well presented and supported by the experimental results. The scintillator application of the Eu-doped composition is significant. A few questions need to be addressed before considering for publication.
1. Change the word ‘battery’ in the introduction and replace it with a simpler word.
2. The red-shifted absorption onset is not explained. Reference 18 is not appropriate and not given the reason behind it. Reference 19 talks about the increasing doping increase absorption onset, is not a proper reason. The authors should put good references and justify properly the redshift.
3. The Transmittance and PLE should be plotted together to make the comparison. Why the 5D0 transition observed in the transmittance is missing in the PLE?
4. Why was the PLE collected at 612nm?
Author Response

(The authors gave the same response as above.)

Round 2
Reviewer 2 Report
The authors have adequately addressed my concerns.
Author Response
Reviewer's comment is "The authors have adequately addressed my concerns". Therefore there is no change and reponse to this comment. Thanks again.